# Extraction, Characterization, and Bioactivity of Phenolic Compounds—A Case on *Hibiscus* Genera

**DOI:** 10.3390/foods12050963

**Published:** 2023-02-24

**Authors:** Carmen Duque-Soto, Xavier Expósito-Almellón, Paula García, María Elsa Pando, Isabel Borrás-Linares, Jesús Lozano-Sánchez

**Affiliations:** 1Department of Food Science and Nutrition, University of Granada, Campus Universitario s/n, 18071 Granada, Spain; 2Departamento de Nutrición, Facultad de Medicina, Universidad de Chile, Santiago 8380453, Chile; 3Department of Analytical Chemistry, Faculty of Sciences, University of Granada, 18071 Granada, Spain

**Keywords:** bioactive compounds, *Hibiscus sabdariffa*, green extraction, HPLC-MS, bioaccessibility

## Abstract

Phenolic compounds have recently gained interest, as they have been related to improvements in health and disease prevention, such as inflammatory intestinal pathologies and obesity. However, their bioactivity may be limited by their instability or low concentration in food matrices and along the gastrointestinal tract once consumed. This has led to the study of technological processing with the aim of optimizing phenolic compounds’ biological properties. In this sense, different extraction systems have been applied to vegetable sources for the purpose of obtaining enriched phenolic extracts such as PLE, MAE, SFE, and UAE. In addition, many in vitro and in vivo studies evaluating the potential mechanisms of these compounds have also been published. This review includes a case study of the *Hibiscus* genera as an interesting source of phenolic compounds. The main goal of this work is to describe: (a) phenolic compound extraction by designs of experiments (DoEs) applied to conventional and advanced systems; (b) the influence of the extraction system on the phenolic composition and, consequently, on the bioactive properties of these extracts; and (c) bioaccessibility and bioactivity evaluation of *Hibiscus* phenolic extracts. The results have pointed out that the most used DoEs were based on response surface methodologies (RSM), mainly the Box–Behnken design (BBD) and central composite design (CCD). The chemical composition of the optimized enriched extracts showed an abundance of flavonoids, as well as anthocyanins and phenolic acids. In vitro and in vivo studies have highlighted their potent bioactivity, with particular emphasis on obesity and related disorders. This scientific evidence establishes the *Hibiscus* genera as an interesting source of phytochemicals with demonstrated bioactive potential for the development of functional foods. Nevertheless, future investigations are needed to evaluate the recovery of the phenolic compounds of the *Hibiscus* genera with remarkable bioaccessibility and bioactivity.

## 1. Introduction

In recent decades, the population’s growing interest into the improvement of health through more natural approaches has inspired research into traditional remedies as sources of molecules with bioactive properties. In this sense, plants have traditionally been used for more than nutritional purposes for hundreds of years, as their medicinal use has been widely recorded and linked with their chemical composition [1,2,3]. Thus, they arise as interesting sources of biologically active compounds with incredible potential as bioactive ingredients.

Research has attributed these broad medicinal effects to the presence of several compounds, such as minerals, vitamins, dietary fiber, and other secondary plant metabolites [4]. Among these substances, phenolic compounds, considered the largest group of plant secondary metabolites, are presented as some of the most interesting bioactive compounds. These molecules have been studied for their extensive bioactive properties, such as antioxidant [5,6], antimicrobial [7,8], anti-inflammatory [9,10,11], and antihypertensive [12,13] effects, among others, which have a great potential for the pharmaceutical, cosmetic, and food industries. 

The phenolic content and its abundance may vary between different plant sources. In this sense, multiple plants have been studied for their phenolic composition. *Hibiscus sabdariffa*, commonly known as roselle, is a tropical edible plant which has been widely used as a food colorant due to the presence of anthocyanins of a bright red color, as well as in the pharmaceutical and cosmetic industries due to its great potential as a source of other phenolic compounds. In this sense, anthocyanins, phenolic acids, and flavonoids are the major bioactive compounds found in this vegetable source [14,15]. In fact, an abundance of bioactive properties found in these substances has been reported, such as antioxidant and antimicrobial activities [16,17], which are related to their proven efficacy in the improvement of pathologies such as obesity [18], hypertension [19,20], and cancer [21,22].

Nevertheless, it should be taken into account that these bioactive effects are dependent on multiple factors. Firstly, these phytochemicals are usually found in plants in low concentrations. This fact implies that an extraction of these compounds is necessary for their use as sources of bioactive compounds to achieve effective doses. These extraction processes would increase the purity of these compounds while reducing the presence of other plant constituents which may interfere with its desired properties. Additionally, for food application purposes, the adequacy of the process for later human consumption must be considered. Hence, green extraction technologies with GRAS (Generally Recognized As Safe) solvents have been developed as desirable alternatives to bioactive ingredient extraction. Thus, ultrasound-assisted extraction (UAE), microwave-assisted extraction (MAE), pressurized liquid extraction (PLE), and supercritical fluid extraction (SFE) have been proposed as the most efficient and convenient processes due to their multiple advantages. The most common solvents are water, ethanol, and their mixtures, together with CO_2_ in the case of SFE. However, those techniques are influenced by diverse instrumental variables that should be addressed for the efficient extraction of phytochemicals. For this purpose, the experimental conditions for an enhanced extraction process must be thoroughly researched through different optimization strategies [23]. The importance of the applied methodology to the obtained phenolic composition is revealed in the observed differences in the bioactivity of extracts obtained using different extraction methods. These results highlight the need for comparison studies that put into light the most convenient extraction systems for each purpose [24,25,26].

In addition, bioactivity is dependent on the ability of these compounds or their metabolites to be absorbed through the intestinal barrier and to reach target sites where they exert their bioactivity, being, therefore, defined by their bioaccessibility [27]. Phenolic compounds are highly sensitive to external conditions, such as pH, presence of oxygen, light, and high temperatures, conditions in which they have proven to be rapidly metabolized. Considering that the enriched phenolic extracts are usually administered in oral formulations, these compounds are submitted to adverse gastrointestinal conditions, which directly affect their stability and observed bioaccessibility [28]. Therefore, the behavior of phenolic compounds in the absorption sites with these adverse conditions should be studied in detail prior to their incorporation into pharmaceutical or food products.

Therefore, the aim of the present review is to compile and discuss the most recent advances in the extraction techniques of *Hibiscus* plants, paying special attention to the applied optimization methodology. Moreover, the influence of the extraction system on the phenolic composition and, consequently, on the bioactive properties of these extracts, including the bioaccessibility as a defining factor of their bioactive potential, is discussed. For this purpose, studies on the extraction of phenolic compounds, analysis of their phenolic profile, related biological activities, and the bioaccessibility of *Hibiscus sabdariffa* performed in the last lustrum have been recorded and evaluated. As far as we are concerned, a comparison of the most recent research focused on *Hibiscus* genera extracts obtained with green methodologies, considering their involvement in phenolic composition, bioactivity, and bioaccessibility, has not yet been performed. 

## 2. Materials and Methods

This systematic review was conducted according to the Preferred Reporting Items for Systematic Reviews and Meta-Analyses (PRISMA) guidelines, 2020 [29]. A comprehensive search of the electronic databases Pubmed and Scopus was performed by the authors for the selection of papers published before May 2022. Searches were made using five combinations. The first combination had the terms “*Hibiscus*” and “Green extraction” and “polyphenols”. The second combination had the terms “*Hibiscus*” and “polyphenols” and “obesity”. The third combination had the terms “*Hibiscus*” and “polyphenols”. The fourth combination had the terms “*Hibiscus*” and “assay” and, the fifth combination had the terms “*Hibiscus*” and “microbiota” and “polyphenols”. The terms of the three last combinations were searched only if they appeared in titles. Furthermore, relevant articles referenced in the selected papers were searched manually, and the same eligibility criteria described above were applied. The identified potential articles were included if they reported data on green extraction methods used in *Hibiscus sabdariffa* samples. In addition, in this search, the selected eligibility criteria were the following: experimental studies, including green extraction methods, which were published from 2016 to 2022. Review articles were excluded from analysis.

The study selection process is depicted in Figure 1. Following the described search strategy, 172 records were identified, of which 46 were excluded as duplicates. Finally, after applying the inclusion and exclusion criteria, 28 studies were found to be eligible for their inclusion in the present systematic review, considering no additional articles identified from the reference lists.

## 3. Results and Discussion

### 3.1. Extraction and Analytical Characterization of Phenolic Compounds from Hibiscus

As previously described, phenolic compounds from *Hibiscus sabdariffa* present incredible industrial potential, with multiple bioactive properties being associated with their phenolic composition. However, the extraction of polyphenols has been reported as a limiting factor for an optimal industrial implementation due to their lower concentration in the starting material. In this sense, different extraction techniques and optimized instrumental conditions should be applied to obtain enriched phenolic extracts. These extraction methodologies would determine the resulting phenolic profiles in the obtained extract and, therefore, their observed bioactivity. Thus, regarding *Hibiscus* samples, previous scientific literature has focused on obtaining enriched phenolic extracts from this plant source using different techniques, as well as reporting their analytical characterization and potential use though different bioactive assays. Their exerted bioactivity is considered as a tool for the optimization of the extraction methodologies that should be applied to obtain the desired phenolic extract for each application.

Thus, the most influential parameters from each study of the considered literature are described in this section. The optimization of multiple instrumental variables is complex, and commonly requires specific strategies that must be planned and performed for an efficient process with reduced costs and tedious experimental procedures. As an alternative to a trial-and-error procedure, experimental design has proven to be a great statistical tool for increasing extraction efficiency with a reduction in the number of tested conditions, thanks to careful previous planning of the experiments. Due to its efficiency, it was applied in most of the reported studies. In general, the experimental designs of the selected literature were focused on the use of response surface methodology (RSM), considering from two to three independent variables coded at three levels, and second-order models. From the variety of design models applied, Box–Behnken (BB) with a factorial design of 3^3^ or 2^3^ [30,31,32] and central composite design (CCD) with a factorial design of 2^3^ [33,34] were the most popular. However, central composite rotable design (CCRD) [35] and face-centered central composite design (FCCCD) [36] were also considered.

In the consulted bibliography, several similarities were found for the selected parameters considering the specific extraction methodology and the modifiable extraction conditions of each study. The main extraction variables to be optimized were temperature, pressure, time, solvent composition, and sample amount, among others. With the objective of efficient extraction of polyphenols and high recovery, most studies selected as response variables the extraction yield of bioactive molecules [34]; individual phytochemical concentrations [32,33]; total phenolic, flavonoid, or anthocyanin contents [30,31,36]; and antioxidant capacity [32]. 

The individual analysis of the phenolic composition by HPLC-DAD/MS of *Hibiscus* extracts stated the prevalence of flavonoids among the sub-classes of phenolic compounds. In this regard, the most widely identified compounds were myricetin, quercetin, and their derivatives; myricetin 3-arabinoside; quercetin 3-glucoside; quercetin 3-rutinoside; and myricetin 3-sambubioside. In addition, phenolic acids were identified in most of the evaluated extracts regardless of the selected extraction methodology. The high abundance of chlorogenic acid, caffeic acid, coumaroylquinic acid, protocatechuic acid, gallic acid, and syringic acid found in the studied extracts should be emphasized. Among the mentioned groups, regarding their bioactive activity, anthocyanins were highlighted among the identified polyphenols, with delphinidin 3-sambubioside and cyanidin 3-sambubioside being the most promising phytochemicals of this medicinal plant. 

The information concerning the extraction technique, experimental design applied, instrumental variables to be optimized, and chemical characterization of the obtained extracts is described in detailed Table 1 for the studied bibliography.

On the one hand, maceration was considered as the traditional extraction technique for obtaining phenolic compounds from *Hibiscus sabdariffa*. In this study, particle size has been identified as a critical target for optimization. The optimum value was in the range of 212–315 µm for particle size of *Hibiscus* material, determined using 2 g of plant powder mixed with 20 mL methanol/water (70:30, *v*/*v*), then stirring at 300 rpm for 24 h at room temperature (18 ± 2 °C). The maximized response variables were TPC and TFC, as determined by UV/Vis spectrophotometry.

As for UAE, for ensuring optimal performance, the studied experimental articles used a multitude of extraction parameters. In this technique, the output amplitude and the extraction time are critical parameters, together with the solvent composition. With regard to the solvent, the % of additional water seems to be critical when Deep Eutectic Solvents (DES) are used in order to decrease the limiting factor of their high viscosity. In Şahin et al., 2020, extraction efficiency was evaluated using a mixture of water with a DES based on citric acid/ethylene glycol, 1/4 M. The output amplitude (15–35%), extraction time (15–45 min), and % of additional water were the tested variables, whereas the response variables selected for optimization were TPC and TAC, analyzed spectrophotometrically. The optimum experimental parameters for a maximum recovery of polyphenols were 32%, 43 min, and 50% water addition, respectively [30]. Moreover, in Zannou et al., 2020, UAE optimization was focused on the use of other DES, concretely composed of sodium acetate:formic acid. The target variables for optimization were the molarity ratio of DES components, the additional water % in DES, and the solvent to solid ratio. The optimum results for TPC, TAC, and TFC (selected response variables) were found, with no additional water in DES composed of a 1:3.6 sodium acetate:formic acid molar ratio and a 10 mL solvent ratio [32]. In addition, in Calliari et al., 2020, the extraction conditions considered for optimization were the output amplitude (%), the extraction time (min), and the solvent composition (water–ethanol, *v*/*v*), whereas the optimized variables were TPC and TAC. Optimization was achieved by extracting for 60 min with 80% ethanol and 80% of output amplitude [31]. Nevertheless, in these studies, the optimization was evaluated through response variables measured by UV-Vis spectroscopy which, as further described in this section, may present limitations due to being a non-selective analytical technique.

When MAE was applied, the evaluated parameters were similar between studies: solvent composition (% EtOH in aqueous mixtures), extraction time, and temperature [31,34]. However, the resulting optimum values for the mentioned parameters differed across the bibliographic results, especially for solvent composition and extraction time. In this regard, while the temperature was very similar between the results (150 and 164 °C), the amount of EtOH in aqueous mixtures varied from 80 to 60%, and the extraction times from 60 to 22 min. These discrepancies may be related to the different analytical techniques applied for the characterization of phenolic compounds (UV-Vis spectrophotometry vs. HPLC-MS), as well as the range of the experimental parameters in the mentioned studies. Therefore, the selected analytical methodologies, UV-Vis spectroscopy and HPLC-ESI-TOF-MS, presented different levels of sensitivity and selectivity for phenolic compound characterization; consequently, this analysis may have an influence on the observed results.

The use of supercritical CO_2_ extraction for obtaining *Hibiscus* phenolic extracts has also been reported, where temperature, extraction pressure, and co-solvent concentration % EtOH) were the optimized conditions. In this study, the selected response variable was the individual phenolic compounds’ concentration determined by HPLC-ESI-QTOF-MS. The optimum instrumental parameters were found to be 50 °C, 250 bar, and 16.7% EtOH [33].

Finally, the optimized parameters for PLE differed between the selected studies. In Pimentel-Moral et al., 2020, the considered conditions were temperature and solvent composition in ethanolic–aqueous mixtures, which have implications for the modification of the dielectric constant of the solvent mixture [35]. Meanwhile, in Şahin et al., 2020, extraction time and particle size of the starting *Hibiscus* sample were also considered as selected parameters for optimization, while the temperature was not considered as critical [36]. In the first study, the optimum conditions were 200 °C and 100% EtOH, for a maximum concentration of individual phenolic compounds determined by HPLC-ESI-QTOF-MS. On the other hand, in the second study performed by Şahin et al., 2021, 80% EtOH, 60 min, and 0.5 mm of particle size reported to be the optimal conditions for enhancing the TPC and TAC, determined spectrophotometrically, in the *Hibiscus* sample.

As previously stated, optimization of these extraction procedures is focused on the increase in extraction efficiency of phenolic compounds. Hence, in order to evaluate the effects of different values and conditions on this yield, an analytical characterization focused on the identification and quantification of bioactive molecules must be carried out. For this purpose, TPC was obtained through the Folin–Ciocalteu method by spectrophotometric measures. The optimum values of TPC in these studies ranged from 20.4 mg GAE/g extract [31] to 233.20 mg of GAE/g extract for UAE [32]. Considering the selected studies and TPC as the response variable, the most efficient extraction method would be UAE. Nevertheless, it must be taken into consideration that other studies reported a low level of phenolic extraction on UAE extracts under optimized conditions compared to other techniques [32]. As mentioned previously, this could be related to the nature of the analytical methodology used to determine this content. This puts into perspective the strength of the presented conclusion, as well as the need for another approach in order to clarify its adequacy. PLE and MAE also exhibited great polyphenolic extraction potential, and, therefore, may also be considered as adequate extraction techniques for the recovery of phenolic compounds from *Hibiscus* plant materials.

For flavonoid analysis, the selected analytical techniques were the following: UV-Vis spectrophotometry [32,37] and HPLC-ESI-TOF-MS [33,34,35]. The TFC analyses achieved values from 10.42 mg ECE/g to 30.5 mg RE/g DW [37] when using UV-Vis spectrophotometry with epigallocatechin and rutin standards, respectively. Pimentel et al. applied an optimized HPLC-ESI-TOF-MS methodology for the quantification of extracts obtained by different extraction methods, resulting in 6.1 mg/g extract for PLE [35], 0.61 mg/g extract for SFE [33], and 14.4251 mg/g extract for MAE [34]. As could be observed, the presented data are very complex, and no tendency could be concluded from those results. Nevertheless, the highest level of flavonoid extraction was achieved when maceration was selected as the extraction method [37]. However, the low selectivity of the selected analytical technique applied in this research must also be considered, as UV-Vis spectrophotometry was used. This fact could influence the selection of an optimal extraction technique and ideal conditions for flavonoid recovery. This finding notwithstanding, PLE and MAE achieved high recovery levels of flavonoids, as was found using the same analytical platform, while SFE seemed to be non-feasible for the efficient extraction of these compounds. In conclusion, MAE surged as the most adequate for the obtention of flavonoids from *Hibiscus sabdariffa*.

Finally, for the analysis of anthocyanin in the obtained extracts, UV-Vis spectrophotometry with 550 nm as the maximum absorption wavelength was the methodology of the selected studies [30,31,32,36,36]. In Şahin et al., 2020, the results concerning the anthocyanin recovery for UAE were 4 mg/g dry weight, while in Deli et al., 2019, the results for UAE were 0.546 g/g extract [30,37]. Moreover, Şahin et al., 2021, obtained a maximum content of 29 mg of cianidin-3-glucoside/g dry weight [36]; whereas Zannou et al., 2020, reported a value of 10.62 mg of delphynidin-3-sambubioside/g dry weight [32]. Regarding these compounds, the most effective extraction methodology was achieved by Şahin et al., 2021, where PLE-optimized extracts were obtained [36]. In Calliari et al., 2020, two extraction techniques (UEA and MAE) reported a value of 509 mg of malvidin equivalents/mL extract for UAE; meanwhile, this value was halved when MAE was used [31]. This indicates that, as previously observed for TPC determination, UAE has proven to be the most effective methodology for anthocyanin extraction from a *Hibiscus* matrix. Nonetheless, these conclusions may be taken with certain reservations, as the determinant values were obtained through non-selective analytical evaluation, and more specific methodologies, such as HPLC systems, were not considered. In fact, this selective analytical methodology was considered in other studies which evaluated the efficiency of PLE, CO_2_-SFE, and MAE techniques [33,34,35]. In these studies, SFE and MAE extracts were analyzed by HPLC-ESI-TOF-MS, while PLE extracts were characterized using HPLC-ESI-QTOF-MS. In this case, anthocyanins were only identified in PLE extractions [35], as SFE did not consider their analysis [33], and in MAE, a degradation of these compounds was observed as a result of extreme extraction conditions [34]. From the selected articles, it can be concluded that PLE is the most optimal extraction method for anthocyanins, even despite the slight degradation that was observed when a high extraction temperature and increased time was applied. 

Lastly, the extraction of phenolic acids from *Hibiscus sabdariffa* was also considered for optimization with different extraction techniques [33,34,35]. The principal identified compounds were neochlorogenic acid, chlorogenic acid, caffeoylshikimic acid, and coumarouylquinic acid. Of all the considered extraction techniques, SFE proved to be the least efficient, as extraction of these phenolic acids was quite limited. This was also observed for caffeoylshikimic acid and cumaroilquinic acid when PLE was the selected technique for extraction. On the contrary, the extraction yield of chlorogenic and neochlorogenic acid was higher with PLE. This could be related to the nature of the optimized extraction conditions for PLE, which seem not to be adequate for the extraction of the former, but more focused on the extraction of the latter. Thus, a better recovery of all the mentioned phenolic acids was achieved when optimized conditions for MAE were evaluated [34]. The combination of different response variables, such as high recovery of total phenolic compounds, flavonoids, and anthocyanins, and considering the maximum TPC, TFC, and TAC, could lead to optimized conditions for the selected technique, making it the most promising for a more complete and efficient extraction of phenolic compounds.

As a conclusion, UAE exhibited the best TPC, followed by PLE and SFE. Nevertheless, PLE appears to be the most adequate technique for recovering flavonoids, anthocyanins, and phenolic acids from *Hibiscus* plants. In addition, it must be highlighted that the analytical technique used for characterization is another crucial parameter to be considered for the optimization of the polyphenols’ extraction efficiency.

### 3.2. Bioactivity Evaluation of Hibiscus Phenolic Compounds

In vitro and in vivo studies have been considered in order to evaluate the bioactivity of *Hibiscus* phenolic compounds. In this sense, most of the studies on the target compounds have focused on the treatment or prevention of obesity (Table 2). Through in vitro studies, adipocytes have been selected as cell models to study this pathology. In this sense, the potential effect of phenolic-rich *Hibiscus* extracts and some metabolites to alleviate obesity-related metabolic complications was evaluated in a 3T3-L1 adipocyte culture [38]. The capacity of these metabolites to activate AMP-activated protein kinases (AMPK) and to modulate lipid metabolism and mitochondrial function was included in this study. These results pointed out that quercetin and its derivatives exhibited effects in the expression and activity of proteins related to energetic and lipidic metabolism. In addition, the mentioned compounds showed a regulating activity on lipid metabolism through the induction of its catabolism. These authors postulated that these compounds increase PPARα expression, a central regulator of lipid metabolism. 

The potential beneficial effects of *Hibiscus* phenolic compounds on obesity through microbiota modulation have also been demonstrated by in vitro evaluation. Indeed, the study of different aspects of gut microbiota has gained importance, as it has proved to be a central element of obesity. Therefore, an increase in *Firmicutes* spp. can be observed in people with obesity, as these microorganisms are related to a better energetic exploitation of food, resulting in an alteration of the *Firmicutes*/*Bacteroidetes* ratio. In this way, the in vitro study presented by Silva et al., 2022 investigated the impact of Metabolaid^®^, a polyphenol-enriched supplement composed of hibiscus (*Hibiscus sabdariffa*) and lemon verbena (*Lippia citriodora*), on gut microbiota composition and functionality. This research was performed by fermentations with human fecal microbiota under gut-simulated conditions [39]. Phenolic extract administration resulted in a modulation of the microbial profile. The promoting changes in fecal microbiota included a significant increase in *Bifidobacterium* and the *Firmicutes*/*Bacteroidetes* ratio. Furthermore, changes in the presence of *Prevotella* and *Akkermansia* were also observed. The populations of these genera, which have been recently associated with obesity, were reduced in the treatment group as compared with the control, in which their presence remained stable. Thus, these authors found that administration of phenolic-rich hibiscus extracts exhibited an effect that could aid in the prevention of obesity through modulation of the microbiota composition. These bioactive properties were related to the presence of both anthocyanins and phenolic acids, and have been further reported through in vitro dynamic studies, as thoroughly described in the following section [40].

Concerning in vivo obesity studies, both animal models and human clinical trials have been described in the literature. With regards to animal models, studies high fat–sugar diet-induced obese rats and hamsters were proposed to evaluate the potential effects of polyphenol-rich *Hibiscus* extracts on obesity and related biochemical parameters. The administration of *Hibiscus* phenolic compounds to Wistar Kyoto rats showed a marked reduction in weight gain and oxidative stress [41]. Lipidic peroxidation was also reduced by the administered extract. This effect was proven to be a result of the restoration of reduced glutathione levels (GSH) and superoxide dismutase (SOD) by phenolic acids and flavonoids. Furthermore, a dose-dependent anti-obesity effect of *Hibiscus sabdariffa* extract was also reported in hamsters [42].

*Hibiscus* extracts have also been evaluated by in vivo human interventions to ameliorate obesity-induced metabolic disturbances. Their application was studied in a double blind, placebo-controlled, and randomized trial on obese and overweight subjects [44]. The combined administration with *Lippia citriodora* phenolic compounds identified significant improvements in body weight, abdominal circumference of overweight subjects, and body fat percentages. The *Hibiscus* phytochemicals were also related to appetite regulation. Administration of phenolic rich hibiscus extracts to an overweight and obese grade-I population resulted in an improvement in the regulation of obesity through an increase in the secretion of hormones and intestinal peptides related to appetite (such as GLP-1, PYY, and ghrelin) [45,46].

In vitro and in vivo approaches have also been used to study the bioactive effects of hibiscus on other pathologies (Table 3). As related to in vitro assays, neuroprotection by *Hibiscus* phenolic compounds has been evaluated as an important potential therapy for multiple sclerosis [47]. Human and rat neuron cultures were exposed to ferrous iron and *Hibiscus sabdariffa* phenolic compounds. These authors highlighted that anthocyanin prevented iron neurotoxicity. Additionally, hydroquinone, naringenic acid, parahidroxibenzaldehide, fumaric acid, and vainillinic acid from hibiscus improved neuro-inflammation in a SH-S5Y5 neuronal cell line [48].

With respect to atherosclerosis, *Hibiscus* phenolic compounds have shown in vitro modulation of the proliferation and migration of vascular smooth muscle cells (VSMCs). Indeed, Chou and co-workers reported that these *Hibiscus* phytochemicals abolished TNF-α-induced metalloproteinase (MMP-9) expression and cell migration by inhibiting the protein kinase PKB pathway [49]. This research confirmed the in vitro results by an in vivo assay developed with New Zealand white male rabbits. Huang et al., 2016, examined the preventive effect of Hibiscus phenolic compounds on insulin resistance linked to epithelial–mesenchymal transition. They concluded that these compounds could be adjuvants to prevent diabetic nephropathy (HK-2 kidney in vitro culture) [51]. Furthermore, hibiscus extracts exhibited an inhibitory effect on colon carcinoma [52,53] and antiviral activity against HSV-2 clinical isolates through the inhibition of VHS replication [54].

Animal in vivo studies have also proven the potential bioactivity of these extracts. *Hibiscus* extracts have been found to have an effect on the improvement of diabetes. Administration of a phenolic-rich *Hibiscus* extract demonstrated an improvement of cardiac function in a type-I diabetes rat model [50]. A significant reduction in coronary flow and lower pressure on the left ventricle were observed, improving cardiac contractility and relaxation rate. In addition, this extract reduced the presence of reactive oxygen species, thus preventing the development of other related pathologies, such as obesity. This bioactivity, as well as a cardioprotective effect, were attributed to the composition of quercetin, rutin, and two main anthocyanins: cyanidin-3-glucoside and cyanidin-3-O-glucosil-rutinoside. An additional study was carried out on the same rat model, where phenolic compounds such as myricetin, syringic acid, ferulic acid, ellagic acid, caffeic acid, and cinnamic acid modulated diabetic-related signal pathways, lowered stress, and improved obesity prevention [43]. Male Wistar rats were used as animal models for evaluating antioxidant effects of different dosages of phenolic compounds from *Hibiscus sabdariffa* [55]. The anthocyanin content of hibiscus extracts were also shown to protect against deleterious effects of lead intoxication in the livers and kidneys of Wistar rats [53]. Lastly, polyphenolic compounds from Hibiscus sabdariffa extract produced an inhibition effect on colon carcinoma metastasis in Balb/c-nude rats [52].

### 3.3. Bioaccessibility and Microbiota Interaction of Phenolic Compounds from Hibiscus

The bioactivity of phenolic compounds, and, more specifically, *Hibiscus* phenolic extracts, as well as their impact on multiple pathologies, was summarized in the previous section. However, these implications are highly dependent on the ability of phenolic extracts to reach specific areas of action. Once polyphenols are released from the food matrix, their instability under gastrointestinal conditions (pH or enzymatic activity) greatly reduces their bioaccessibility and bioavailability, thus impairing the achievement of their desired activities. In this way, a wide range of factors affects phenolic stability, such as chemical structure, interaction with other compounds present in the food matrix, presence of precursors, biological conditions, the nature of the absorption process, or, more importantly, even the interaction with the existing microbiota. 

As far as Hibiscus phenolic compounds are concerned, a small number of studies have conducted research to obtain a better understanding of their complex and high bioaccessibility under simulated gastrointestinal digestion. 

The standardized digestion protocol Infogest was applied to establish the upper gastrointestinal digestion of *Hibiscus* phenolic extracts [56]. High bioaccessibility of phenolic acids, flavonoids, anthocyanins, and related metabolites was reported. These compounds are abundant in pre-digested hibiscus extracts and, more importantly, have been identified as responsible for most their biological activities. 

Dynamic models have also been used for evaluating the gastrointestinal digestion of these phytochemicals, allowing simulation of the full gastrointestinal tract from stomach to colon [39,40]. In these studies, colonic microbiota mediated phenolic bioconversion of Hibiscus extracts. As has previously been described, phenolic extract administration resulted in a modulation of the microbial profile. These investigations suggested that these phenolic compounds showed antimicrobial activity, microbiota modulation, and, more specifically, obesity-related microbiota effects (phenolic acids, anthocyanins). Indeed, the dynamic in vitro human gastrointestinal simulations reported that phenolic acids, flavonoids, and anthocyanins were able to have a modulating effect on colonic microbiota composition. Prior to in vitro fermentation, 45 bacterial genera were identified, with post-fermentation results presenting significant variations in 10 of them. Mainly, a high recount of *Bifidobacteirum, Bacteroides, Catenibacterium,* and *Lactobacillus* were observed. In this sense, the increase in *Bacteroides* was inhibited after 24 h digestion due to the antimicrobial activity of *Hibiscus sabdariffa*. Thus, phenolic compounds from its extract were able to promote the genera *Bifidobacteirum* and *Lactobacillus* while significantly reducing the presence of *Bacteroides*. The anthocyanin and phenolic acid contents in hibiscus extracts were able to modulate microbiota, with a significant impact on obesity. 

Thus, as can be observed, although the bioaccessibility of hibiscus extracts has not been widely researched, these extracts have proven to be high in significant bioactive phenolic compounds, reported phenolic acids, flavonoids, and anthocyanins. These bioaccessible hibiscus compounds have also been proven to have a modulating effect on colonic microbiota, which poses them as potential sources of compounds for improving gut health.

## 4. Conclusions

*Hibiscus sabdariffa* has proven to be a promising plant source of an abundance of phenolic compounds, from flavonoids such as myricetin, quercetin, and derivatives; phenolic acids, especially caffeic, chlorogenic, coumaroylquinic, gallic, and ellagic acids; and anthocyanins, such as delphinidin, cyanidin, and derivatives. The best extraction methodology for maximum total polyphenol recovery was UAE, whereas PLE was the most adequate technique for recovering flavonoids, anthocyanins, and phenolic acids. In fact, PLE reported a high extraction yield for all studied compounds; thus, it should be proposed for a combined extraction from *Hibiscus* polyphenols. On the contrary, CO_2_-SFE seemed to be the least efficient for obtaining enriched phenolic extracts from *Hibiscus*. Moreover, the extraction of these compounds appears to be essential for their bioactivity, as their effects on recorded pathologies such as obesity are related to their content in the mentioned compounds. A combination of optimization strategies using response surface methodology, along with the use of advanced analytical platforms such as HPLC-MS, is proposed as an optimal strategy for future research, as the enhanced selectivity of this analytical platform would improve the optimization of phenolic compound extraction, thus reducing potential disturbing factors and variations between studies. It is important to remark that in spite of the bioactivity of the Hibiscus phenolic compounds being demonstrated in both in vitro and in vivo assays, more research is still needed in this field for a deeper understanding of their high and complex bioaccessibility. 

## Figures and Tables

**Figure 1 foods-12-00963-f001:**
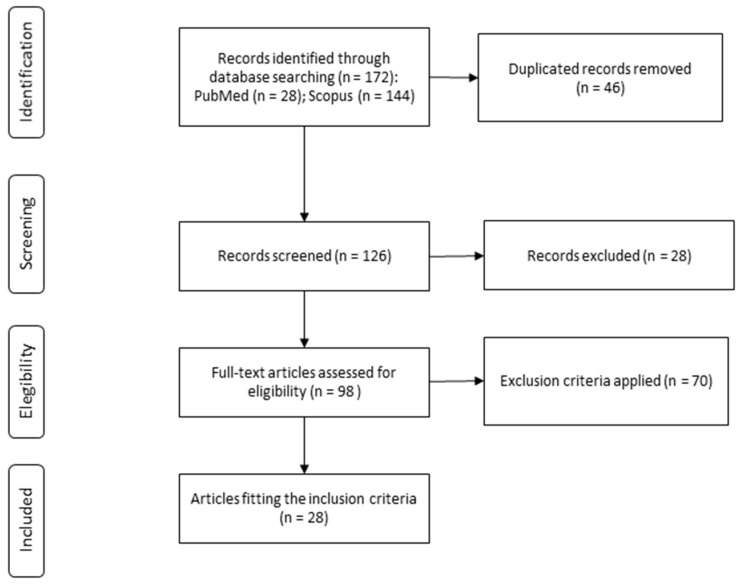
Study selection process for the systematic review.

**Table 1 foods-12-00963-t001:** Extraction methods, optimal conditions, and analysis of bioactive compounds for the obtention of *Hibiscus sabdariffa* extracts.

Extraction Technique	Experimental Design	Extraction Parameters	Optimum Values	Response Variable	Analytical Technique	References
**Maceration**	No experimental design	Sample particle size (mm): 0–180, 180–212, 212–315, ≥315 µm, unsieved	212–315 µm	TPC TFC	UV/visible spectrophotometry (Folin–Ciocalteu method, absorbance recorded at 510 nm)	[37]
**UAE**	RSM (BB)	Output amplitude (15–35%), extraction time (15–45 min), additional water % (15–25%, *v*/*v*) in DES solvent (citric acid:ethylene glycol 1/4 molar ratio)	32%, 43 min, 50% water addition	TPC TAC	UV/visible spectrophotometry (Folin–Ciocalteu and pH differential methods)	[30]
**UAE**	No experimental design	Output amplitude (%), extraction time (min), solvent composition (water–ethanol, *v*/*v*)	80%, 60 min, ethanol 80% (*v*/*v*) aqueous solution	TPC TAC	UV/visible spectrophotometry (Folin–Ciocalteu method and colorimetric assay)	[31]
**UAE**	RSM (3-level BB)	Molarity ratio of DES components (sodium acetate:formic acid, 1, 2.5, and 4 molar ratio); additional water % in DES (0, 30, 60%); solvent to solid ratio (10:0.5, 25:0.5, 40:0.5 mL/g)	1:3.6 sodium acetate:formic acid molar ratio; 0% additional water and 10 mL solvent ratio	TPC TAC TFC	UV/visible spectrophotometry (Folin–Ciocalteu method, pH differential method, absorbance at 510 nm)	[32]
**MAE**	No experimental design	Temperature (°C), % EtOH in aqueous mixtures extraction time (min)	150 °C, 80% EtOH, 60 min	TPC TAC	UV/visible spectrophotometry (Folin–Ciocalteu method and colorimetric assay)	[31]
**MAE**	RSM (2^3^ CCD)	Temperature (50–150 °C), % EtOH in aqueous mixtures (15–75%) and extraction time (5–20 min)	164 °C, 59.63% EtOH, 22.2 min	Extraction yield (%)	HPLC-ESI-QTOF-MS (quantification of individual phenolic compounds)	[34]
**SFE**	RSM (2^3^ CCD)	Temperature (40, 50, 60 °C), pressure (150, 250, 350 bar) and co-solvent composition (7, 11, and 15% EtOH)	50 °C, 250 bar, and 16.7% EtOH	TPC (individual phytochemical concentrations)	HPLC-ESI-QTOF-MS	[33]
**PLE**	RSM (CCRD)	Temperature (40–200 °C) and solvent composition (EtOH-H_2_O, 0–100% *v*/*v*)	200 °C and 100% EtOH	TPC (individual phytochemical concentrations)	HPLC-ESI-QTOF-MS	[35]
**PLE**	RSM (FCCCD)	% EtOH in aqueous mixtures (10–80%), extraction time (40–60 min) and particle size of *Hibiscus sabdariffa* sample (0.5–2 mm).	80% EtOH, 60 min, and 0.5 mm	TPC TAC	UV/visible spectrophotometry (Folin–Ciocalteu and pH differential methods)	[36]

RSM: response surface methodology. BB: Box–Behnken. CCD: central composite design. CCRD: central composite rotable design. FCCCD: face-centered central composite design. TPC: total phenolic content. TAC: total anthocyanin content. TFC: total flavonoid content. DES: deep eutectic solvent. As can be observed in Table 1, for obtaining optimal hibiscus phenolic extracts, different extraction methodologies have been reported in previous literature. Among the evaluated techniques in the reviewed articles, the most promising seem to be maceration, ultrasound-assisted extraction (UAE), microwave-assisted extraction (MAE), CO_2_-supercritical fluid extraction (CO_2_-SFE) with the use of ethanol as co-solvent, and pressurized liquid extraction (PLE). A great variety of parameters have been proposed as determinants for the extraction of polyphenols from this plant.

**Table 2 foods-12-00963-t002:** Bioactivity of phenolic compounds from *Hibiscus genera* extracts on obesity.

Assay Type	Model		Bioactive Effect	Responsible Compounds	Reference
In vitro	3T3-L1 pre-adipocyte culture		Reduction in metabolic stress	Quercetin derivatives	[38]
	Static fecal fermentations (SIMGI^®^)		Modulation of obesity-associated microbiota	Anthocyanins, phenolic acids	[39]
In vivo (animal)	Wistar Kyoto rats		Oxidative stress and weight gain reduction	Phenolic acids, flavonoids	[41]
	Hamsters		Lipogenesis and adipogenesis inhibitionin pre-adipocytes	Total polyphenols, flavonoids	[42]
	Albino Sprague–Dawley rats		Diabetic stress signaling modulation	Myricetin, syringic acid, ferulic acid, ellagic acid, caffeic acid, cinnamic acid	[43]
In vivo (humans)	Overweight women (ages 36–69)	Experimental group	Reduction in intracellular triglycerides in hypertrophied adipocytes	Delphinidin-3-O-sambubioside	[44]
		Control group		Cyanidim-3-O-sambubioside	
	Grade-I obese subjects (ages 18–65)	Experimental group	Appetite regulation in overweight and grade-I obese population	Sambubioside derivates	[45]
		Control group			
	Overweight women (ages 30–75)	Experimental group	Appetite regulation in overweight population	Anthocyanins	[46]

**Table 3 foods-12-00963-t003:** Bioactivity of phenolic compounds from *Hibiscus sabdariffa* extracts on other pathologies.

Study Target	Assay Type	Used Model	Bioactive Effect	Responsible Compounds	Reference
Neuroprotection	In vitro	Human and rat neuron cell cultures	Prevention of iron neurotoxicity	Anthocyanins	[47]
		SH-SY5Y cell culture	Improvement of neuro-inflammation	Hidroquinone, naringenic acid, parahidroxibenzaldehide, vinilic acid, fumaric acid	[48]
Cardioprotection	In vitro	Bioresource Collection and Research Center rat cell culture	Anti-aterosclerofic effect	Total polyphenolic content	[49]
	In vivo	Sprague–Dawley rats	Cardioprotector effect on type-I diabetic rats	Cyanidin-3-glucoside, cyanidin-3-O-glucosil-rutinoside, quercetin, rutin	[50]
Mesenquimatic–epithelial transition	In vitro	HK-2 cell culture	Prevention of renal mesenquimatic epithelial transition	Caffeic acid, chlorogenic acid, gallic acid	[51]
	In vivo (animals)	Sprague–Dawley rats	Prevention of renal mesenquimatic epithelial transition	Caffeic acid, chlorogenic acid, gallic acid	[51]
Colon protection	In vitro	DLD-1 human colon cancer cell culture	Inhibition of colon carcinoma metastasis	Total polyphenolic content	[52]
	In vivo (animals)	Balb/c-nude rats	Inhibition of colon carcinoma metastasis	Total polyphenolic content	[52]
Liver and Kidney Toxicity	In vivo (animals)	Wistar rats	Reduction in lead toxicity in liver and kidney	Anthocyanins	[53]
Antioxidant activity	In vivo (animals)	Wistar rats	Antioxidant effect	Delphinidin-3-O-sambubioside and cyanidin-3-O-sambubioside	[54]
Antiviral activity	In vitro	Vero cell culture and clinical isolates of HSV-2	Antiviric effect of HSV-2	Protocatechuic acid	[55]

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
