# Peer review of "Extraction, Characterization, and Bioactivity of Phenolic Compounds—A Case on Hibiscus Genera"

_foods, 2023, doi:10.3390/foods12050963_

Round 1
Reviewer 1 Report
The paper represents a sound and thorough review dealing with the extraction, chemical and physiological characterization of phenolics from Hibiscus genera.
The terms and the research criteria have been conveniently described and defined in view of giving a good representation of the research from the last 6 years in this field.
Despite the papers matching the criteria were only 28, other publications have been taken into account for comparison purposes and to give a complete overview of subjects such has extraction methods or physiological efficacy of phenolic form Hibiscus.
Some awkward sentences have been found from time to time, which I need should impose a further revision of the spelling.
In all, the paper is suitable to be processed further in my opinion, giving valuable, up to dated and complete information on the subject.
Author Response
We thank the reviewer for this positive evaluation. A spelling and grammar revision of the article has been carried out for a better understanding.
Reviewer 2 Report
The review article entitled "Extraction, characterization, and bioactivity of phenolic compounds: a case study on Hibiscus genera" is quite an interesting piece of work. It provides a summarized insight into the most important plant-derived bioactive molecules, mainly the wide group of phenols. The selection of literature sources and multiplicity of presented extraction methods have good scientific soundness.
However, the title and introduction, as well as other sections of the manuscript, suggest the involvement of optimization methods. It is a bit misguiding because the authors do not mention much about the optimization methods involved. I recommend that the authors add one small subchapter regarding the optimization methods and models applied by the studied scientific papers, such as: if it was an in-line optimization or off-line, maybe by DoE (design of experiments) or QbD (quality by design) or even RSM (response surface methodology). Authors may even extend the tables in the paper with an additional column explaining optimization methods.
Author Response
We thank the reviewer for this positive evaluation.
In accordance with this comment, a new information has been included in the manuscript, concretely in Results and Discussion “3.1. Extraction and analytical characterization of phenolic compounds from Hibiscus”. This information describes the optimization methods and models applied by the studied scientific papers. Moreover, table 1 has been modified for including the information regarding the experimental design applied in each paper.
Reviewer 3 Report
The work entitled "Extraction, characterization, and bioactivity of phenolic compounds – a case on Hibiscus genera" was reviewed and comments and observations were made in order to improve its quality. This manuscript is really a very good prepared manuscript and contains serious consideration as potential publication. I think the work will contribute to the existing literature and should be processed further after a minor revision:
1) Abstract (Line 12-25): The Abstract must be explained with more details
2) Keywords (Line 26): Those words included in the title should be removed from this section.
3) Line 70-75: The text at the end of introduction must be revised. The motivation of the paper is not sufficiently justified in the introduction.
4) Introduction: Even if the paper aim has been properly evidenced, it is strongly suggested to highlight the originality and added value of the present work with respect to the Literature about the same topic.
5) The novelty should be highlighted in abstract, introduction and conclusions. Also the application or the advantages of this technology regarding the existing ones should be discussed
6) In Materials and Methods, all experiments, equipment and condition must be described in detail.
7) Materials and methods: Authors should provide more detailed information on the Hibiscus. The place of purchase, time of harvest, variety, and so on.
8) Results and discussion section: The results have been well represented but the discussion needs to be improved (i.e. comparison to other works and citations ).
9) Conclusion: Please make a few-line conclusion about the work. It can be the essence of all the results. It should be suggestive about the best practice for future works.
Author Response
Reviewer 3
Comments and Suggestions for Authors
The work entitled "Extraction, characterization, and bioactivity of phenolic compounds – a case on Hibiscus genera" was reviewed and comments and observations were made in order to improve its quality. This manuscript is really a very good prepared manuscript and contains serious consideration as potential publication. I think the work will contribute to the existing literature and should be processed further after a minor revision:
We thank the reviewer for this positive evaluation. The authors have addressed all the reviewer´s suggestions point by point.
1) Abstract (Line 12-25): The Abstract must be explained with more details
The abstract have been modified with a more detailed description.
2) Keywords (Line 26): Those words included in the title should be removed from this section.
The keywords have been modified according to this comment.
3) Line 70-75: The text at the end of introduction must be revised. The motivation of the paper is not sufficiently justified in the introduction. 4) Introduction: Even if the paper aim has been properly evidenced, it is strongly suggested to highlight the originality and added value of the present work with respect to the Literature about the same topic.
We thank the reviewer for his/her suggestions for improving the introduction section. According to these comments, the introduction has been revised and modified to clearly justified the motivation of the paper and also the originality and novelty of the present work respect to the available literature.
5) The novelty should be highlighted in abstract, introduction and conclusions. Also the application or the advantages of this technology regarding the existing ones should be discussed
We agree with the reviewer in this point. The novelty of the present manuscript has been highlighted along the document, especially in abstract, introduction and conclusions sections.
6) In Materials and Methods, all experiments, equipment and condition must be described in detail. 7) Materials and methods: Authors should provide more detailed information on the Hibiscus. The place of purchase, time of harvest, variety, and so on.
In this sense, regarding the nature of systematic review of the present manuscript, the details regarding the Hibiscus material, the equipment, experiments and instrumental conditions of each analysed study have not been described in this section. The reason for this decision is the limited extension of the manuscript, due to this information would considerably increase the number of pages of the document. Nevertheless, it should be taken into account that the critical information concerning all these aspects have been described in results and discussion section and properly referenced.
8) Results and discussion section: The results have been well represented but the discussion needs to be improved (i.e. comparison to other works and citations).
Sections 3.1 and 3.2 have been rewritten for a better understanding. New information has been added to the discussion section in order to improve the quality of the manuscript.
9) Conclusion: Please make a few-line conclusion about the work. It can be the essence of all the results. It should be suggestive about the best practice for future works.
We have incorporated the reviewer´s suggestion to improve the conclusion remarks.
Reviewer 4 Report
The article is a good fit for the journal. However, the part I would like to see more improvements in is the introduction. There is a strong lack of some important previous references. Please add the following:
Zamora, R.S., Baldelli, A. and Pratap-Singh, A., 2023. Characterization of selected dietary fibers microparticles and application of the optimized formulation as a fat replacer in hazelnut spreads. Food Research International, p.112466.
Mesgaran, M.B., Mashhadi, H.R., Zand, E. and Alizadeh, H.M., 2007. Comparison of three methodologies for efficient seed extraction in studies of soil weed seedbanks. Weed Research, 47(6), pp.472-478.
Benayad, Z., Martinez-Villaluenga, C., Frias, J., Gomez-Cordoves, C. and Es-Safi, N.E., 2014. Phenolic composition, antioxidant and anti-inflammatory activities of extracts from Moroccan Opuntia ficus-indica flowers obtained by different extraction methods. Industrial crops and products, 62, pp.412-420.
Ammar, I., Ennouri, M. and Attia, H., 2015. Phenolic content and antioxidant activity of cactus (Opuntia ficus-indica L.) flowers are modified according to the extraction method. Industrial crops and products, 64, pp.97-104.
Author Response
We thank the reviewer for the positive evaluation and the important contribution for improving the manuscript. The introduction has been modified and the references has been introduced and discussed.